# HIV Vaccine Development: 35 Years of Experimenting in the Funding of Biomedical Research

**DOI:** 10.3390/v12121469

**Published:** 2020-12-19

**Authors:** Stuart Z. Shapiro

**Affiliations:** Vaccine Research Program, Division of AIDS, National Institute of Allergy and Infectious Diseases, 5601 Fishers Lane, Rm 9C20B, Bethesda, MD 20892-9829, USA; sshapiro@niaid.nih.gov; Tel.: +1-240-292-6155; Fax: +1-240-627-3109

**Keywords:** HIV-1, HIV vaccines, vaccinology, vaccine design, biomedical research funding

## Abstract

Funding vaccine development research is more complicated than simply putting out an announcement of funds available. The funders must decide whether product development can be accomplished by purely applied research, or whether more fundamental knowledge is needed before product development can be started. If additional basic knowledge is needed, identifying the specific area of the knowledge gap can be a challenge. Additionally, when there appears to be a clear path of applied research sometimes obstacles are encountered that require a return to more basic work. After deciding on the work to be done, funders must attract the scientists with the broad range of needed skills to cover all the stages of development. Collaborations must be promoted and alliances with other funders and industry must be developed. Funders use multiple tools and strategies to accomplish these tasks with varying success.

## 1. Introduction

The rapidity with which vaccine developers have pivoted to working on COVID-19 vaccine development may have given many the impression that all that needs to be done to facilitate vaccine development for serious diseases is for funding agencies to announce the availability of funding. The several decades experience of funding HIV vaccine development demonstrates that it is not always that simple. Sometimes, the path to vaccine development appears to be clear. However, the market for some vaccines is so uncertain that the industry is not motivated to develop a vaccine, while academic scientists see an easy development path as too little a scientific challenge, or they do not even apply for funding because they fear the absence of a basic research challenge means grant applications will not be well received by peer reviewers. At the other extreme, the path to vaccine development is so unclear that the industry will not start the work until academic or government scientists have established the way. And often few academic researchers know where to start. In this situation, the challenge for the funders is to recruit the best scientists, with expertise in the newest technologies, to work in a field that may not promise quick results. In almost all cases, proper vaccine development, even when straightforward, is too large and complex a project to be easily divided into two-year to five-year grant programs, such as the standard grants available at the US National Institutes of Health (NIH). Furthermore, vaccine development is a complex process, which is beyond the capability of a single scientist. Before a vaccine candidate can be tested in people, it must be manufactured to current good manufacturing practice (cGMP) standards. Then, it must be tested in clinical trials for immunogenicity, safety and efficacy. Both activities involve different skill sets from those possessed by the vaccine product designers. Thus, building collaborations is extremely important. Funding agencies have developed many different tools to help vaccine developers get over these hurdles, and all of them have been used in the effort to develop an HIV vaccine over the past three decades. This article, written from the perspective of a Program Officer’s experiences in the Division of AIDS (DAIDS) of the National Institute of Allergy and Infectious Diseases (NIAID) at the US National Institutes of Health (NIH), will provide some detail about the tools NIAID employs, why and how they were applied in HIV vaccine development and to what extent they were useful.

## 2. HIV Vaccine Development Challenges

Vaccine development for most infectious diseases is applied research (at the end of World War II, presidential science adviser, Vannevar Bush, laid out a conceptual framework for thinking about science and technology in which he drew a clear distinction between pure or basic research and applied research [1]). It starts with the observation of individuals who have been infected and subsequently recovered from disease by their own immune responses. If those curative responses could be induced by a vaccine before the individual is exposed to the pathogen the impact of the disease would be lessened or prevented completely. Thus, the analysis of immune responses of recovered individuals establishes the possibility of developing a vaccine guided by the quality, antigen or epitope specificity and quantity or titer of those responses. The vaccine developer then applies a small number of well-established, practical strategies (whole-killed, attenuated pathogen and surface protein or toxin subunit methodologies which have been refined and improved by decades of basic research [2] and, more recently, added to by the strategy of vector-based vaccines, thanks in no small part to HIV vaccine development funding [3]) to the task of developing a vaccine. Unfortunately for HIV vaccine development, no humans have cured themselves of HIV-1 infection. Additionally, HIV vaccine developers have long been hindered by the fact that HIV-1 does not replicate in any small animal model, and simian immunodeficiency virus (SIV) models do not allow direct testing of HIV vaccines. Despite these critical hurdles to standard vaccine development HIV vaccine developers started where most others start, with virus envelope protein vaccine candidates intended to induce neutralizing antibody responses to the only viral protein expressed on the virion surface. Quickly scientists came to understand that the extreme mutability of HIV-1 led to an enormous variety in virus envelopes to which the uninfected individual is exposed, likely precluding vaccine development based on a feasible number of envelope types. Thus, HIV vaccine development presents the challenge of inducing an unknown amount (or titer) of an unknown immune response to an unimaginable diversity of antigens without an animal model for preclinical testing.

Clearly, the type of research required is beyond the usual “applied” research of vaccinology focused directly on vaccine product development. Neither is it the classical pure “basic” research that seeks to understand fundamental scientific problems without thought of immediate practical applications (the type of research performed with many unsolicited R01 NIH grant awards). What is required is a type of research that falls between basic and applied research, a research that looks to enhance basic knowledge to rapidly apply it to a practical problem. This is not a startlingly new concept at the NIH, as many biomedical scientists perform “basic” research essentially to apply their results to some specific disease. Such work has been called targeted basic research, purposive basic research, mission-oriented basic research, use-inspired basic research or, more recently, Pasteur’s quadrant research (see Donald E. Stokes, “Pasteur’s Quadrant: Basic Science and Technological Innovation” [4]. Stokes divides research into four quadrants based on its relationship to (a) the quest for fundamental understanding and (b) considerations for use. Research that is essentially a quest for fundamental understanding without consideration of use is pure basic research similar to the physics research of Niels Bohr. Research that utilizes already existing fundamental knowledge to create new products or processes for use is purely applied research similar to that conducted by Thomas Edison. Research that seeks to extend the frontiers of understanding but is also inspired by considerations of use defines Louis Pasteur’s quadrant; research enhancing basic knowledge that has immediate utility). Fundamental yet use-inspired research has had a natural home at the NIH with the great expansion of Government funding for science following World War II. However, there is not a large cadre of scientists trained to intentionally perform such work to quickly apply it, so it usually requires building coalitions/collaborations of scientists with different skills and original interests, and with the latest novel, cutting edge technologies (in recognition of this need, a new institute within NIH, the National Center for Advancing Translational Sciences (NCATS; [5]) was established in 2011. NCATS’ mission is to catalyze the generation of innovative methods and technologies that will enhance the development, testing and implementation of diagnostics and therapeutics across a wide range of human diseases and conditions. However, mostly it has focused on therapeutics development, and vaccine developers have yet to access it fully).

Additionally, it is not always clear what scientific disciplines or specific basic research will provide the answers needed. It requires broad but pragmatic, original thinking. Finding, interesting and then recruiting and teaming for a long-term effort such scientists of high quality is one of the funders’ challenges. 

In addition to obtaining funding, there are other hurdles that hinder scientists moving into a new and challenging field. These include access to clinical samples and to specialized research materials, standardized and reference reagents. Also helpful is the knowledge that there will be support/assistance for costly activities outside the investigators immediate area of expertise, such as nonhuman primate challenge studies, cGMP manufacturing, regulatory agency-required toxicology studies and clinical trials.

## 3. The Tools in the Funders’ Toolbox

NIH Extramural research awards are essentially of two main types: grants and contracts. Grants are considered “enabling” mechanisms whereby the Government enables the funded investigator to perform the research the investigator proposes and directs. Contracts are “acquisition” mechanisms whereby the Government purchases the research the Government wants performed and Government administrators more closely control the direction of the research. A third type of funding mechanism is called a cooperative agreement. Cooperative agreements are grants and thus “enabling” mechanisms for the researcher to perform the research that the researcher has proposed; however, they are used when substantial Government administrative scientist involvement is anticipated between the awarding Institute and grantee because the Government has a direct practical interest in the results of the research. There are many different types of grants, varying in the size of the award (smaller awards usually requiring fewer preliminary data to justify), the term of the award (usually two to five years but sometimes up to seven years) and the complexity of the award (e.g., in some awards, the research is directed by a single investigator, while others are to teams of scientists under single principal investigator leadership or led by a collaborating team of two or more scientists). In some cases, the NIH has funded even larger efforts as centers or consortia. The diversity of grant types is described on the NIH website [6]. Another tool in the funders’ toolbox is to supplement existing grants to perform work of the same type yet over and above the research proposed in the original application; this is sometimes a faster way to exploit new scientific opportunities. A special category of grants (and sometimes research contracts) is the Small Business Innovative Research/Small Business Technology Transfer (SBIR/STTR) program; these are grants to small businesses and small businesses allied with academic investigators that fund applied research more directly aimed at product development than the basic research NIH grants to academic scientists.

All NIH grants and contracts are assigned to a Program Officer (called a Contract Officer’s Representative, COR, in the case of contracts). Program Officers are unique to the US NIH; many other national biomedical research funding and philanthropic research funding organizations do not have such staff. Program Officers are a special cadre of administrators trained first as scientists (most have PhDs and/or MDs) who follow the progress of the research to offer advice and guidance. This entails alerting investigators to special administrative requirements that could hinder the work if not attended to, asking probing questions, suggesting collaborations to bring in needed helpful technologies and trouble-shooting problems they have seen solved in other awards. As a cadre that is closely following the research in a specific field, Program Officers are also well placed to recommend redirecting funds into areas with immediate promise for results that can be quickly applied because of newly developed research technologies and areas needing special attention because specific hurdles to important product/process development have been identified. This is done by writing funding opportunity announcements (FOAs) and obtaining bureaucratic approval to set aside funds for applications and proposals that propose to perform research to achieve the goals of the FOA. Another tool in the funders’ toolbox is the ability to fund repositories of clinical samples, specialized reagents, and databases; these are usually best funded with contracts because the task to be done is clear, so Government supervision ensures it is done and accessible to all investigators.

A tool that deserves special mention is the organization of workshops and meetings on specialized topics. Workshops accomplish several goals at the same time. They bring together investigators active in a field to present their latest findings, share ideas and assess progress. They are also used to introduce new technologies and basic research findings from outside the field to investigators working in a field, and to encourage collaborations between established investigators and those with the new concepts and technologies. Lastly, they help Program Officers identify scientific gaps and determine whether there is enough interest and the possibility of promising science to justify set-aside funds for future FOAs. These functions are also frequently served by the Scientific Advisory Boards NIH organizes to provide annual review of progress in the larger (program project and center or consortium awards) grant and research contract awards.

Lastly, an important tool is alliances. Alliances can be with other funding organizations. The first funding alliance that the NIAID Division of AIDS (DAIDS) Vaccine Research Program had was with the International AIDS Vaccine Initiative (IAVI) founded in 1994. The alliance with IAVI continues, however, since IAVI established a research facility, there have been established some procedures restricting the free flow of information to prevent IAVI scientists from having an unfair advantage in access to Federal funds, while continuing to facilitate an equitable distribution of research dollars. Additionally a long and productive alliance with the Bill and Melinda Gates Foundation has been formed. In both cases, we have each assisted our grantees by referring them to the other funders when our mechanisms were not the best fit for the specific research challenge. We have frequently consulted with each other and shared valuable insights into progress as well as scientific gaps in the field. In addition, since 2003, DAIDS is part of a broad international HIV vaccine funder alliance called the Global HIV Vaccine Enterprise. DAIDS, also, has longstanding research alliances with other research parts of the US Government, such as the NIAID intramural Vaccine Research Center (VRC) and the US Military HIV Research Program (USMHRP). Alliances can also be with industry partners. Such industry alliances may be essential to get vaccine candidates properly manufactured and to perform clinical trials; and they are necessary to prepare for eventual commercialization and distribution of a successful vaccine. An alliance with community organizations can also be incredibly useful. One valuable lesson learned from HIV vaccine development is the importance of community engagement. The rapid pace of antiretroviral drug development would not have occurred absent the push for mobilization of resources from the HIV infection at-risk community of men who have sex with men (MSM) population in the United States (especially important were ACT UP, the AIDS Coalition to Unleash Power; and TAG, the Treatment Action Group). HIV vaccine development has also benefited from this mobilization of resources and with an alliance with AVAC, the AIDS Vaccine Advocacy Coalition. The importance of community in vaccine development is not a new story. In the 1940s and 1950s, the effort at that time to develop a polio vaccine was similarly spurred by community engagement—the March of Dimes [7]. The bigger the scientific challenge, the more certain it is that a long-term commitment will be needed, which will only benefit from alliances and dedicated community support.

## 4. Evolving Use of Funders’ Tools

HIV vaccine development has not been a linear process; it has fluctuated between very applied research and more mission-oriented basic research, as well as between antibody-based approaches and T-cell-based approaches. The changing use of funding tools reflects (and sometimes precedes) developments in the field. Additionally, while individual investigators must decide on a best approach to focus the work of their laboratory, a large funding organization is wise to pursue several different approaches at the same time, because in vaccine development for a significant disease success of any approach is more important to public health than success of any specific approach. Thus, this section cannot fit into a strictly linear chronology; a timeline, which illustrates the overlapping nature of funding strategies, is provided to help the reader (Figure 1).

In the beginning of HIV vaccine development, the effort was left almost entirely to the private sector, because it was perceived to be essentially applied research for which a market for the product was assured. Thus, the first products were pursued by small start-up biotech companies largely with private funding: MicroGeneSys worked on a subunit (gp160 envelope protein) preventive vaccine candidate, VaxSyn [8]; and a company formed by Dr. Jonas Salk worked on a “whole-killed” product, Remmune, to be used as a therapeutic vaccine [9]. These products progressed as far as Phase 2 clinical trials, but by the mid-1990s, both were seen to have failed. In the early 1990s, two gp120 envelope protein candidate HIV vaccines were also under development by other private companies (Chiron and Genentech) [10]. They sought funding from the NIH for large scale, Phase 3 (efficacy) clinical testing, but by then, many scientists had lost hope in the ability of a simple subunit protein vaccine to be effective, and the NIH declined to fund advanced trials of these products.

While these envelope protein and whole-killed virus vaccine approaches were being pursued, it had already become clear to many academic scientists that the usual approaches to vaccine development pursued by industry would not produce an effective HIV vaccine. Thus, in the late 1980s NIAID, whose focus until that time had largely been on thought-to-be more scientifically challenging anti-retroviral drug development, became involved in HIV vaccine development funding. The recognition that HIV vaccine development would require multiple skills and resources beyond the scope of a standard NIH R01 award led to the release in 1988 of a Request for Applications (RFA, the old term for a FOA for grants) for the National Cooperative Vaccine Development Groups for AIDS (NCVDG) awards. The NCVDG awards were program project grants, supporting multiple, linked R01-sized projects with supporting core facilities to create teams of collaborating scientists to perform basic research through vaccine product development/manufacture research. This RFA was released several years running and sponsored an annual meeting, which all funded investigators attended to share results. Another strategic mechanism started in the 1990s and continued to this day has been the release of FOAs for small grants (in the beginning R21s but progressing to R01 funding levels) for innovative, high-risk, high-reward research not requiring preliminary data to facilitate funding of novel ideas. In 1993, an RFA was released to promote R01 (single project) research into mucosal immunology to promote HIV vaccine development. In later years, FOAs were released to promote HIV-related research in basic vaccine discovery, innate immunity, B-cell immunology, B-cell help and the microbiome.

While academic scientists were funded to design new candidate HIV vaccines and perform preclinical development research, it was clear that the clinical testing would require more resources than could be provided by even the larger program project grants. So, in 1988, DAIDS established a clinical trial apparatus. The AIDS Vaccine Evaluation Group (AVEG) comprised several AIDS Vaccine Evaluation Units (AVEUs), a Data Coordinating and Analysis Center, a Central Immunology Laboratory, a Mucosal Immunology Laboratory and a Viral Immunology Laboratory. AVEG activities were conducted under contract with active coordination by DAIDS. Another contract apparatus was set up by DAIDS to perform nonhuman primate challenge studies to test candidate HIV vaccines, the Simian Vaccine Evaluation Units (SVEUs). Additionally established were the AIDS Research Reagent [11] and Specimen repositories [12], as well as the Los Alamos-based HIV Database and Analysis Unit [13], which continue to this day as valuable resources to enable the research.

Nevertheless, HIV vaccine development progressed slowly. In 1996, a high-level review panel (the Levine committee) made several important recommendations for changes to how the NIH was managing HIV/AIDS research funds [14]. For vaccine development, the most important recommendation was that the NIH put more effort into understanding the interaction of HIV with the immune system and to make NIH-sponsored AIDS vaccine research “more vigorous and effective”, characterized by some at the time as a call to go “back to basics” [15]. Specifically, the panel recommended a major effort to get capable extramural basic scientists (especially immunologists) engaged in HIV vaccine research and to do this by convening workshops with expert immunologists as well as doubling the portion of AIDS vaccine research funding going to R01 grants. The panel also advised decreasing the portion of HIV vaccine development research managed by NIH staff; one way in which this was accomplished was by changing the HIV vaccine clinical trials apparatus from a contract into a cooperative agreement (the HIV Vaccine Trials Network, HVTN, launched in 2000) thus more directly enlisting the research expertise and insight of academic clinical investigators.

Some clinical scientists proposed that preventive HIV vaccine development follow the iterative exploratory clinical trial pathway successful in cancer chemotherapy development. This is a mostly applied research approach (it has been referred to as the “empirical” approach [16]) rather than a mission-oriented basic research approach; it is perhaps representative of the tug-of-war in HIV vaccine development research between these two different types of science. In this strategy, multiple successive trials are used to refine treatment regimens, producing regimens with greater and greater efficacy. Instead of simply refining vaccine regimens, HIV vaccine developers would use iterative trials to inform the discovery of immune responses that correlate with protection to serve as a guide for vaccine improvement. In the long run, this strategy may be impractical because of cost and the time required for the necessary large-scale clinical trials for a preventive vaccine [17]; however, absent many alternatives, it has been pursued. An additional, immediately pragmatic problem is that the strategy relies heavily on the manufacture of vaccine candidates in compliance with regulatory agency expectations for clinical trial products (cGMP); this is not a skill available in most academic laboratories and was not found among the investigators funded under the NCVDG program. In the late 1990s, the DAIDS Program developed two new funding mechanisms to partner academic scientists with industry to enable vaccine candidate product manufacture. At the same time, the more basic research part of the NCVDG program was replaced with program project awards not expected to engage in manufacturing, the HIV Vaccine Research and Design program (HIVRAD), which continues to this day. Both new product development mechanisms supplied comparable amounts of funds over a 5-year award period; however, one used a cooperative agreement mechanism (the Integrated Clinical/Preclinical AIDS Vaccine Development grants; IPCAVD) and the other used a contract mechanism (the HIV Vaccine Design and Development Teams contracts; HVDDT). The contract mechanism was used for the HVDDT program to attract proposals from industry because it was thought that, while academic scientists were used to working with government grants, industry was more accustomed to government contracts. HVDDT contracts were awarded yearly for 5 years, while IPCAVD grants continue to be awarded almost yearly to this day. Additionally, DAIDS developed a contract mechanism (the Preclinical Master contract) to task contract manufacturing organizations (CMOs) to perform cGMP manufacturing as well as specific manufacturing tasks (e.g., production cell bank preparation, analytics, fill/finish, stability studies) for investigators with a promising vaccine candidate but lacking all the funds required for complete manufacturing. Additionally, as it became recognized that manufacturing HIV-1 envelope proteins presented special problems for manufacturers, a series of annual workshops were organized to share problems and solutions among product designers and manufacturers. These mechanisms and efforts, together, have facilitated the entry of more than 20 different modalities of candidate HIV vaccines into clinical trials. However, few have reached the stage of efficacy testing and only one is still in efficacy testing today (an Ad26-vectored GagPolEnv vaccine boosted with gp140 envelope proteins in clinical trials HVTN 705 and HVTN 706) [18].

For a while, the “empirical” strategy appeared to be working. In 2009, the USMHRP and Thailand Ministry of Public Health co-sponsored RV144 trial demonstrated modest efficacy (31%) in prevention of the acquisition of HIV infection with a vaccine regimen that induced little if any neutralizing antibodies or cytotoxic T-cell activity [19]. The candidate products were redesigned for use in South Africa based on a correlates analysis of the modest protection observed in RV144 [20], and a large funder alliance was established, the Pox Protein Public Private Partnership (P5) [21], to perform a Phase 2B proof of concept clinical trial (HVTN 702). However, in 2019, this trial was stopped, as it failed to demonstrate any efficacy [22]; that has dampened enthusiasm for the empirical strategy.

By the late 1990s, most HIV vaccine scientists had given up on the idea of an antibody-inducing, simple subunit envelope protein vaccine composed of a small number of variant HIV-1 envelopes. However, the alternative for some was not use-inspired basic research; rather, it was another applied research approach with a similarly small component vaccine designed to induce cytotoxic T-cell responses. A major effort was undertaken by an alliance between DAIDS and Merck Co. to perform a proof-of-concept Phase 2B efficacy trial of Merck’s CD8 CTL-inducing HIV vaccine (the STEP/Phambili trials). Not only did this vaccine product fail to protect against infection, but it may have instead enhanced the risk of infection in some subgroups of individuals receiving the product [23]. After these results were announced, NIAID convened an HIV Vaccine Research and Development Summit in Rockville, MD on 25 March 2008 to review the NIAID HIV vaccine research portfolio and to seek input on the appropriate balance between vaccine discovery and development. A clear consensus emerged from the Summit participants that HIV vaccine research should shift in favor of discovery (once again a call for “back to basics”) [24]. However, already three years earlier, wary of over-reliance on the T-cell-based vaccine approach, Program Officers in DAIDS teamed up with Program Officers in the more basic research-focused sister NIAID Division of Allergy, Immunology and Transplantation (DAIT) to stimulate more HIV vaccine mission-oriented basic research into antibody responses. This effort started with administrative supplements to develop collaborations between B-cell immunologists and HIV vaccine scientists, which were followed rapidly by FOAs for R21, R01, cooperative agreement program project grants and meetings/workshops.

At the same time as the extramural program was funding academic and industry scientists to develop candidate HIV vaccine products through DAIDS, the intramural program of the NIH was taking a different, and more purposive basic research approach. A Vaccine Research Center (VRC) was established on the NIH campus in 2000 to perform basic research to establish mechanisms of inducing long-lasting protective immunity against HIV and other pathogens that present special challenges to vaccine development. The founding director of the VRC, Dr. Gary Nabel, took an innovative approach to mission-oriented basic research, which he described in an article in *Science* [25] as research directed more at expanding the knowledge landscape rather than testing hypotheses. Understanding that the accuracy and predictability of a hypothesis depend on the validity of the inputs used to generate and test it, instead of encouraging strictly hypothesis-based research, the VRC hired experts in new technologies and pushed them to dive into HIV immunology, virology and structural biology to expand the knowledge base upon which hypotheses could be formulated. This approach obviates the need to agonize over the specific basic research needed. However, this approach can only be taken by a large, well-funded center.

In 2003, two decades after the discovery of HIV with no vaccine clearly in sight, a group of 24 internationally known scientists came together to publish a policy article calling for a “Global HIV Vaccine Enterprise” [26]. Among other things (e.g., this article also provided the impetus for the formation of the international funder alliance called the Global HIV Vaccine Enterprise mentioned earlier), they called for the establishment of multiple HIV vaccine development centers, self-contained like the VRC or virtual networks of collaborating scientists at different institutions or commercial entities that would both compete and collaborate to advance HIV vaccine development. NIAID supported this call by launching a very large cooperative agreement-funded extramural research center version of the intramural VRC, the Center for HIV/AIDS Vaccine Immunology (CHAVI). At the start, CHAVI was directed to focus on purposive basic research into why the early host immune responses were unable to control HIV infection rather than more applied vaccine development tasks so that the vaccine candidates eventually developed by CHAVI would represent new thinking rather than minor improvements on old products. The inspiration for this was the studies of Dr. Bernard Fields into reovirus pathogenesis in order to illuminate new targets for antiviral intervention [27]. Early on, attention was focused on both T-cell and B-cell (antibody) responses, but gradually, the development of broadly neutralizing antibodies (BNAbs) became the major interest. At the same time, IAVI was encouraging independent investigators to collaborate more closely on BNAbs in the Neutralizing Antibody Center (NAC). Soon after the NIAID funded CHAVI, the Bill and Melinda Gates Foundation put a similar amount of funds into HIV development grants that were networked together in a collaboration called the Collaboration for AIDS Vaccine Discovery (CAVD); many of these awards also focused on the induction of BNAbs. Quickly, it was discovered that the main obstacle to induction of BNAbs against HIV-1 was a requirement for extensive somatic hypermutation of the antibodies from an unfortunately very small number of germline antibody genes that could even begin to recognize the rapidly mutating envelope surface protein. This was a fundamental shift in understanding what was needed to make an HIV vaccine that was derived from mission-oriented basic research. How to actually induce a BNAb response may still be as much a basic research problem as an applied research problem; however, for now, the field is taking a more applied, iterative preclinical and Phase 1 clinical trial approach by testing multiple immunogen constructs for their ability to engage the rare germline antibody genes and select for mutations that promote the required antibody affinity maturation. Neutralizing antibodies have been shown to offer protection, by passive transfer, against SHIV (simian immunodeficiency virus with an HIV-1 envelope protein) infection in nonhuman primates [28], and more recently, against HIV-1 infection in humans (the AMP trial [29]), so the target immune response is clearly predictive of vaccine efficacy, while how to induce it may still be a basic research question. Thus, the induction of BNAbs as the basis of an HIV vaccine is characterized as the “theoretical” approach [16] to distinguish it from the “empirical” approach where the correlate(s) of protection would only be determined after efficacy was seen. In the nomenclature introduced by Stokes, the “empirical” vs. “theoretical” approaches would have been called Edison’s quadrant vs. Pasteur’s quadrant research, respectively [4].

Despite the new promise of the antibody-based approach to developing an HIV vaccine, DAIDS is beginning to work with investigators to reinvigorate T-cell-based vaccine approaches to HIV vaccine development. This is still at the stage of workshops and meetings to explore the gaps in fundamental understanding that hinder applied vaccine development. If specific gaps can be identified and research tools are available that promise new understanding, it may justify funding new mission-oriented basic research on this vaccine approach in the hope of preparing this area for applied research in the future.

## 5. Tactical Challenges for Funders

Before announcing the availability of funding, a task of the funders is to decide what type of research (applied or mission-oriented basic research) must be performed and what type of researchers should be recruited to perform it. Then, constant reassessment is required to determine when a different type of research may be needed. When the applied research/development path is blocked by a gap in fundamental understanding, failure of clinical trials makes this readily apparent. However, then, funders need a strategy for identifying and enabling the mission-oriented basic research to be done, and this is not always easy. NIAID has found several approaches useful for focusing on needed new basic research, including (a) high-level review committees, public meetings, workshops and “summits”; (b) FOAs targeting specific areas for multiple research awards; (c) large centers/consortia to study HIV pathogenesis and early anti-viral immune responses; and (d) a large center to develop the pathogen data landscape.

High-level review committees, public meetings, workshops and “summits” are useful in making assessments that research directions must be supplemented or change. However, the usefulness of open public meetings of scientists for identifying the specifically needed new research is sometimes compromised by individual scientists keeping their best ideas to themselves to gain advantage in their next grant application. Additionally, private, brain-storming meetings to seek specific direction on new funding opportunities are not allowed for public funding agencies who must avoid giving any applicants advance information that will give them an unfair advantage for obtaining public funds. Program Officers in NIAID have found annual external scientific advisory board (SAB) reviews of large grant efforts a useful way to get outside experts to identify gaps and suggest new paths of research because the invited advisers feel encouraged to help their scientific peers working on the grant, and as advisers to the grant, they get a feeling of sharing in the success of the grant. At the same time, the NIAID Program Officer gets useful advice for the entire field. Another way in which the program has been successful in promoting a high-level, think-tank type of collaborative free exchange of ideas has been by giving larger grants that fund multiple investigators who then evolve from competitors to collaborators because the success of the award is the success of all of them; this frequently fuels vibrant scientific discussions.

It was mentioned earlier that another task of the funders is to develop broader collaborations as research becomes more complex and requires more resources to use increasingly sophisticated and expensive technologies. Giving larger grants that fund multiple investigators has been observed in HIV vaccine development to promote the needed broad collaborations. Especially useful for bringing mission-oriented basic research to bear on HIV vaccine development has been the funding of very large centers/consortia such as the VRC and CHAVI. These have been good mechanisms for developing the informational landscape to allow the formulation of better hypotheses for hypothesis-testing research without the need for funders to identify the needed new basic research too specifically. As well as the exposure in these big centers of younger scientists to multiple state-of-the-art technologies and many peers who are potential future collaborators has been a good training experience. However, the funder must be careful to consider how such large efforts can be maintained because recruiting scientists, especially young scientists, to short-lived efforts can be disruptive of both the research and their careers. At the same time, funders should be careful to avoid creating ill will in the field by giving the impression of an unfair disproportionate distribution of resources. This can only be done by maintaining a healthy level of smaller grant funding.

An important task for funders is creating the interest to recruit basic scientists when needed to work in a new field. Meetings and service on advisory committees and on grant review panels helps to foster interest among basic scientists. Additionally, all vaccine scientists benefit from the availability of resources such as specimen, reagent and data repositories. The knowledge that these many resources exist lowers the threshold for scientists to enter a new field. Applied researchers need to be assured that the resources for advanced stages of development will be available (e.g., manufacturing and clinical trials capabilities). Basic researchers need to see an interesting scientific challenge and the possibility of getting publishable results. All benefit from a sense of purpose, energy and excitement in the field as well as the opportunity to collaborate fostered by well-run scientific meetings.

Perhaps the most successful experience recruiting new scientists to the HIV vaccine development that the Division of AIDS has had has been the initiative to apply B-cell immunology. This started as a collaboration between Program Officers in DAIDS and DAIT to target administrative supplements to partner B-cell immunologists with HIV vaccine scientists. FOAs for R21 and R01 grants and U19, program project cooperative agreements followed. Three years after the first grant awards, DAIDS was able to bring all the funded investigators together for a very exciting workshop to discuss and promote this area of interest. This effort, together with the work of the VRC, the IAVI NAC, and the CHAVI and CHAVI-follow up awards has led to a deeper understanding of BNAb development to the point that today, this is the major focus of HIV vaccine development. The point is that this new direction did not just happen—it was nurtured by funders using multiple funding tools at the same time.

## 6. Conclusions

From the outside, the jobs of the funders in vaccine development are simply to find the right scientists to do the work and enable them with funds, access to research resources and a product development and testing infrastructure. Funding agencies have developed many tools to do these jobs and many of those have been described in this article. This is the case when vaccine development requires only applied research. This is where most vaccine development starts, as applied research utilizing the well-developed standard tools and practices of traditional and modern vaccinology. When this works, as appears to be the case for COVID-19, there is no need to do anything else.

However, sometimes the standard vaccine development approaches do not work, and then, the funder has the more difficult task of determining what type of new research is needed. Are there fundamental gaps in knowledge that must be filled by more basic research before vaccine product development can be attempted? The answer to this question may change, as some hurdles are overcome with new technologies or new paths are opened by advances in fundamental understanding. Then, the field can return to an applied research effort. Continuous assessment is necessary to determine when the scientific landscape has changed enough to require new or a different type of research. When new basic research is required, it can be problematic determining what new research is required. Frequently, it is only known specifically what new research was required after critical results are obtained. The funders can take two different approaches: either fund many small research projects in many areas or fund a large research center such as the VRC or CHAVI. Confronted with this dilemma, NIAID chose to do both. It appears to have been successful in that there has been a fundamental change in basic understanding represented by the nature of BNAbs. However, it must be admitted that this may not be enough to make possible an HIV vaccine. The field may have to return to basics yet again. As in research, with funding strategies immediate success is not assured. However, when the task of vaccine development is important for the lives of millions of people, as it is with HIV/AIDS, scientific advance cannot be allowed to stand still; it must be nurtured with all the tools at the funders’ disposal. Nevertheless, it must be admitted that a comprehensive strategy for vaccine development when there is a gap in fundamental understanding can be very complicated and costly. Therefore, promoting communication, building collaborations and establishing alliances, especially with community, are extremely important.

The HIV vaccine development field is not unique. Vaccine development has progressed too slowly in other fields of great public health importance such as tuberculosis and malaria. The hard-won lessons learned in HIV vaccine development may have some applicability for these other challenging public health problems.

## Figures and Tables

**Figure 1 viruses-12-01469-f001:**
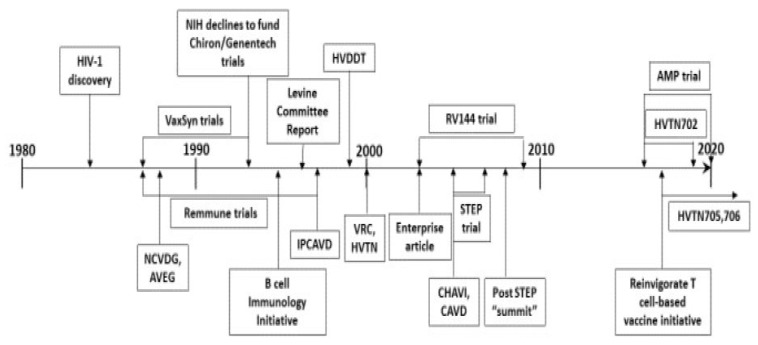
HIV vaccine development timeline landmarks from the funders’ perspective.

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
