# Peer review of "HIV Vaccine Development: 35 Years of Experimenting in the Funding of Biomedical Research"

_viruses, 2020, doi:10.3390/v12121469_

Round 1

Reviewer 1 Report

The article titled “HIV Vaccine Development: 35 years of experimenting in the funding of biomedical research” by Stuart Z. Shapiro is a review written from the perspective of a Program Officer, and provides a detailed overview of different NIH funding mechanisms as applied to the development of a HIV vaccine. It traces what has been a long and arduous path toward the goal of obtaining an effective HIV vaccine, and how funding mechanisms evolved and adapted to new discoveries, challenges, successes and failures in the field. While written in the context of HIV vaccine development, a lot of the information contained in this review - different funding mechanisms and how they are utilized, the discussion on basic and applied research, how funders weigh decisions and what the rationale are to fund or not to - will be generally useful for researchers. The review is well-written and engaging. A few suggestions for the author that may help further improve the readability and organization of this very comprehensive review:

  1. The review is divided into the following sections:
  2. Introduction
  3. HIV Vaccine Development Challenges
  4. The Tools
  5. Evolving Use of Tools
  6. Tactical Challenges
  7. Conclusions

It may help to make some of the section headings more explicit that will give readers an overview of what will be discussed in the section that follows. For example, the section heading “HIV Vaccine Development Challenges” informs the reader what is going to be discussed in the section. Along those lines, -

 “The Tools” could be replaced by “Tools in a Funder’s Toolbox” OR “Tools in the Funders’ Toolbox”

“Evolving Use of Tools” could be replaced by “Evolving use of funding tools”

“Tactical Challenges” could be replaced by “Tactical Challenges in Funding Decisions” OR “Tactical Challenges for Funders”

  1. In this sentence in footer of Page 4: “However, since IAVI established a research facility there have been established some barriers to the free flow of information to prevent IAVI scientists from having an unfair advantage in obtaining Federal funds.”, while the intent of the statement is clear, the notion of funders “establishing barriers” seems to have a somewhat negative ring to it.

Can this be re-phrased, for example “However, since IAVI established a research facility there have been established some procedures restricting the free flow of information to prevent IAVI scientists from having an unfair advantage to Federal funds, and to continue to facilitate equitable distribution of research dollars.”

  1. Section 4: “Evolving Use of Tools” although packed with useful information, is the longest section in the review. The author may consider dividing this into subsections to improve readability.
  2. The author makes references to the success of COVID 19 vaccine development, and categorizes this recent success as an example of applied research. It may be relevant to point out that this application was based on research that was considered basic in the not too distant past, thus providing an example of funders’ decision to support basic research when utility might be unclear but the biological question is compelling.
  3. While the author mentions this in the concluding sections of the review, it may be relevant to emphasize that the years of funding that has gone into the development of an effective HIV vaccine, has generated know-how and infrastructure that has facilitated fight against other diseases.

Author Response

I thank the reviewer for his/her thoughtful reading of my manuscript. I have accepted most of the suggestions and incorporated them into the revised manuscript. Specifically I have changed the section titles to be more informative as recommended in comment #1. I have changed the wording of the reference to establishing new boundaries to information exchange between DAIDS and IAVI as recommended in comment #2. In response to comments # 4 and #5 I have added a reference to the modernizing of vaccinology by Dr. Rino Rappouli and some language attributing this to basic research as well as to the funding going into HIV vaccine development in lines 56-60 of the manuscript. I wholeheartedly agree with the point about the contribution of HIV vaccine research to vaccinology and it was the main point of my last paper (Shapiro, SZ. Lessons for general vaccinology research from attempts to develop and HIV vaccine. 2019 Vaccine 37(26):3400-3408) but I didn't emphasize it in this manuscript because I thought it might detract from the main message about funding tools and challenges which HIV vaccine research shares with many other challenging fields of research. About comment #3, I have not broken down this admittedly long section into subsections because I think that the way it is truly reflects the long and sometimes tortuous history of HIV vaccine development where frequently several paths have been pursued at the same time. It is difficult to break up into a linear story with discrete segments; that is the nature of the work and so structuring the manuscript that way parallels the way it actually happens. 

     I hope the revised manuscript is now acceptable for publication. Thank you again for reviewing it.

Reviewer 2 Report

HIV vaccine development: 35 years of experimenting in the funding of biomedical research.

By Stuart Z. Shapiro.

This review describes nicely the rationale of the funding route towards the development of vaccines, the hurdles and decisions to be made focusing in this paper on HIV vaccine development. The funders task is not only to find the right scientist to do the work and provide them with funds, access to research resources, product development and testing infrastructures as concluded by the author but also to determine what type of new research is needed in case standard vaccine development approaches do not work. As a scientist it is an eye opener to see what is happening and required behind the curtains to provide funding for vaccine development in general and for HIV in particularly.

This review is very well suitable for publication.

One remark: the footnote numbers 1 and 2 are missing at the bottom of the 2nd page.

Author Response

Thank you for your complimentary comments and for appreciating that I was trying to pull back the curtain that funders too often unintentionally hide behind in making funding decisions. Everybody tends to get too stuck in their own silo these days; sometimes the focus helps us get our jobs done but sometimes it causes unnecessary confusion. And thank you for pointing out the missing footnote numbers; they were in the submitted document but appear to have been left out when the document was uploaded to the journal format. I have added them back to the resubmitted document.